



# Land-based wind turbines with flexible rail transportable blades – Part I: Conceptual design and aeroservoelastic performance

Pietro Bortolotti[1], Nick Johnson[1], Nikhar J. Abbas[1,2], Evan Anderson[3], Ernesto Camarena[3], and Joshua Paquette[3]

[1]National Renewable Energy Laboratory, Boulder, CO 80303, USA
[2]University of Colorado Boulder, Boulder, CO 80309, USA
[3]Sandia National Laboratories, Albuquerque, NM 87185, USA

**Correspondence:** Nick Johnson (nick.johnson@nrel.gov)

**Abstract.**

This work investigates the conceptual design and the aeroservoelastic performance of land-based wind turbines whose blades can be transported on rail via controlled bending. The turbines have a nameplate power of 5 MW and a rotor diameter of 206 m, and they aim to represent the next generation of land-based machines. Three upwind designs and two downwind designs are

presented, combining different design goals together with conventional glass and pultruded carbon fiber laminates in the spar caps. The results show that controlled flexing requires a reduction in the flapwise stiffness of the blades, but it represents a promising pathway to increase the size of land-based wind turbine rotors. Given the required stiffness, the rotor can be designed either downwind with standard rotor preconing and nacelle uptilt angles or upwind with higher-than-usual angles. A downwind-specific controller is also presented, featuring a cut-out wind speed reduced to 19 m per second and a pitch-to-

stall shutdown strategy to minimize blade-tip deflections toward the tower. The flexible upwind and downwind rotor designs equipped with pultruded carbon fiber spar caps are found to generate the lowest levelized cost of energy, 2.9% and 1.3%, respectively, less than the segmented design. The paper concludes with several recommendations for future work in the area of large flexible wind turbine rotors.

## 1   Introduction

Wind energy today is one of the most cost-competitive sources of electricity. The recent reduction in levelized cost of energy (LCOE) for wind has been possible thanks to various factors, among which is a continuous increase in turbine size. In terms of cost per kilowatt, larger machines help decrease balance-of-station costs in addition to operation-and-maintenance costs. An increasingly robust supply chain has led to fairly constant values of turbine capital costs per kilowatt, whereas larger rotors

and taller towers increase annual energy production (AEP). The larger rotor-swept areas have also helped to increase power generation at low wind speeds, which is especially attractive in markets that are characterized by a high penetration of wind



energy. Here, electricity price is often inversely proportional to wind speed, which has pushed the technology trends toward larger rotor diameters and lower values of specific power. Low specific power also allows for better predictability of power production, ultimately supporting a higher penetration of renewables in the energy mix (Bolinger et al., 2020).

The increasing size of wind turbines is clearly visible in offshore platforms—for example, the latest International Energy Agency (IEA) Wind Task 37 15-MW reference wind turbine mounts a rotor of 240 m in diameter at a hub height of 150 m above mean sea level (Gaertner et al., 2020). Land-based wind turbines have also been increasing in size, but the trend has been less aggressive than offshore. The main reason behind this difference can be explained by the fairly rigid transportation constraints that land-based installations must meet. A first set of constraints has traditionally impacted the tower size because conventional

steel towers cannot exceed base diameters of 4.5 m for road transport. Recently, however, alternative technologies—such as hybrid concrete-steel towers, field-welded towers, and self-erecting towers—have offered promising pathways to overcome these limits (Dykes et al., 2018). A second set of constraints impacts the design of the blades, with stringent limits on maximum chord, prebend, sweep, and length. The first three quantities are constrained to have the blade fit within a rectangular clearance section. A common value for maximum chord is 4.75 m, and this limit is usually respected by imposing aerodynamically

suboptimal chord values as well as flat-back and truncated airfoils. Prebend is often constrained to 3 or 4 m at the blade tip, and designers increase rotor cone and nacelle tilt angles to reach sufficient blade-tower clearance. For sweep, only a few manufacturers adopt it in their blade designs, and the curvature is always limited to no more than a few meters to limit the growth in blade torsional moment, which impacts the design of the pitch system (Bortolotti et al., 2019b). Although these constraints do not necessarily represent hard stops to the growth in turbine size, the logistical limit on blade length is harder

to address. Logistics companies have broken several barriers during the past years, but the longest monolithic blades currently being transported on land are still limited to 70–75 m. One technological solution to overcome this limitation consists of segmenting the blade into two parts, where the inner part is as long as it can be transported, and the outer part is as long as the joint can support. Although they are promising and commercially viable, segmented blades are not the perfect solution because they suffer more disadvantages than monolithic configurations. First, the joint represents an extra weight that causes

blades to become heavier and more expensive, with example increases of +10-+15% assumed by Griffin et al. (2019). Blades also need to be assembled on-site, with nonnegligible technical challenges to meet required tolerances and avoid damages to the structures. Finally, the long-term reliability of large, jointed, flexible blades is not well known.

    This work aims to investigate an alternative pathway—namely, the controlled flexing during rail transport of monolithic blades. The solution was initially proposed at a conceptual level by Griffin et al. (2019), and it is investigated here in more

detail. A matrix of five wind turbine rotors is defined. Three of the five rotor designs meet the rail transport constraints and are characterized by a much more pronounced flexibility than the other two designs, which represent state-of-the-art technology and are used as a reference. One of the two baseline rotors is segmented, whereas the other is an unconstrained design. Two of the three flexible designs are downwind. Two blade designs have spar caps made of conventional glass fiber-reinforced plastics, whereas three adopt pultruded carbon fiber laminates. All designs go through the same design process and analysis of

the aeroservoelastic performance.



The research activity is part of the Big Adaptive Rotor (BAR) project, which is funded by the U.S. Department of Energy and consists of partners Sandia National Laboratories, the National Renewable Energy Laboratory, Oak Ridge National Laboratory, and Lawrence Berkeley National Laboratory. The project focuses on a 5-MW platform with a specific power of 150 W m$^{-2}$, which leads to a rotor diameter of 206 m and blade lengths of 100 m. This paper focuses on the conceptual design of rotors that can be transported by rail and on their aeroservoelastic performance. A parallel paper presents the detailed 3D finite-element analysis of the flexible blades (Camarena et al., Under Submission at the Journal of Wind Energy Science).

The paper is structured as follows. First, Sect. 2 presents the design workflow, the model simulating the rail transportation of blades, and the databases of materials and airfoils adopted to design the blades. Next, Sect. 3 introduces the numerical framework used to assess the aeroservoelastic performance of the baseline and of the flexible rotors. A scheme of the workflow described in these two sections is reported in Fig. 1. Next, a reference baseline design, a segmented design, and the flexible rotor configurations are presented in Sect. 4. The results of the time domain simulations together with the linearized solutions are discussed in Sect. 5, and Sect. 6 presents the comparison of LCOE for all designs. The paper closes with Sect. 7, which lists the main conclusions and the recommendations for future work.

## 2 Design workflow

The design process developed for this work is implemented in the Wind-Plant Integrated System Design and Engineering Model (WISDEM®) framework (Ning and Petch, 2016; NREL, 2021b). WISDEM is a multifidelity design framework for wind turbines and wind plants implemented in Python. WISDEM is fully open source, and it is based on the multidisciplinary design, analysis, and optimization library OpenMDAO (Gray et al., 2019).

### 2.1 Models

WISDEM is made of several components all wrapped by an OpenMDAO interface. The next subsections introduce the models that have been adopted for this study and distinguishes among rotor, drivetrain, tower, and models for the cost analysis.

#### 2.1.1 Rotor

WISDEM simulates the rotor with steady-state models. The rotor aerodynamics are solved with the blade element momentum model CCBlade (Ning, 2014), the elastic properties of the composite blades are obtained by running the cross-sectional solver PreComp (Bir, 2006), and the deformations are obtained by running the Timoshenko beam solver Frame3DD (Gavin, 2014). A steady-state regulation trajectory is implemented by maximizing the rotor performance until rated power is reached and by imposing a constant power above rated wind speed. An optional constraint on maximum blade tip speed is respected by maximizing power performance while maintaining constant rotor speed. The AEP is computed here, whereas the ultimate loads are estimated by running a CCBlade simulation at rated pitch and rotor speed values and at a wind speed corresponding to the peak of the three-sigma gust for the extreme turbulence model (IEC-61400-1; Ning and Petch, 2016). This approach to estimate loads is known to be somewhat overconservative, but it is capable of capturing the relative trends, and it is suitable to




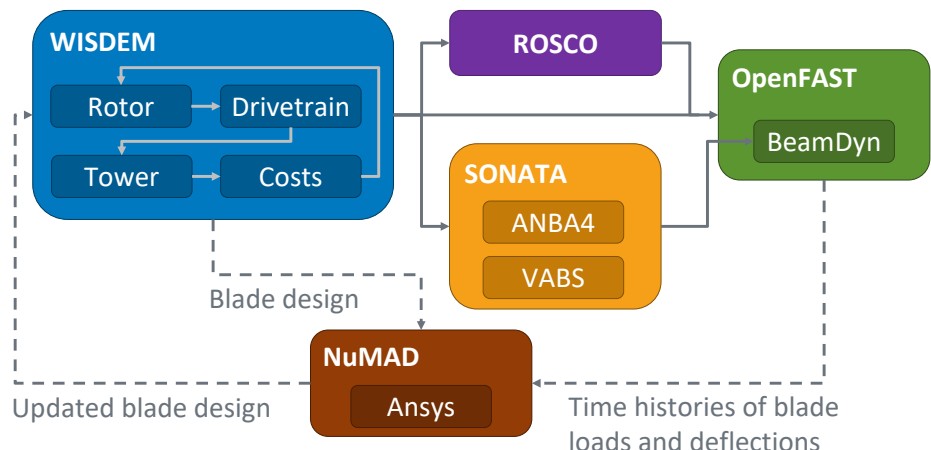

**Figure 1.** Scheme illustrating the sequence of numerical frameworks adopted in this study. WISDEM is described in Sect. 2; and ROSCO, SONATA, and OpenFAST are described in Sect. 3. NuMAD and the data flow to it, which is represented by the dashed line, are described in Camarena et al. (Under Submission at the Journal of Wind Energy Science).

run an iterative optimization loop on standard hardware in only a few minutes, offering the designer the chance to explore a wide solution space. The work described in Camarena et al. (Under Submission at the Journal of Wind Energy Science) shows how the detailed structural design of the blades is clearly necessary, although the simplified loading approach leads to fairly

90    realistic designs that are close to the optimal point.

### 2.1.2   Drivetrain

This work assumes geared wind turbine configurations, and the engineering models described in Guo et al. (2015) are used to design the various nacelle components. Simplified analytical relations are adopted to size the pitch system, the hub, the main bearings, the low-speed shaft, a three-stage gearbox, the high-speed shaft, the generator, the bedplate, the nacelle cover,

95    a transformer, the yaw bearings and motors, and the auxiliary systems. The components are designed assuming a desired value of gearbox ratio and overhang, whereas the loading comes from the rotor models described in Sect. 2.1.1.

### 2.1.3   Tower

The tower is defined as a sequence of conical hollow cylinders made of steel. Within Frame3DD (Gavin, 2014), the tower is modeled as an elastic beam with a point mass at the top. The mass and center of gravity of the rotor nacelle assembly together

100   with the loads at the tower top are fed to the model, which computes values of maximum stresses, natural frequencies, and buckling limits. The ratio of wall thickness to outer diameter and the rate of change of wall thickness along the tower height are also computed and can be constrained during the design optimizations.





| Quantity | Value |
| --- | --- |
| Number of turbines in the plant (-) | 120 |
| Balance-of-station costs ($ kW$^{-1}$) | 459 |
| Operational expenditure costs ($ kW$^{-1}$) | 44 |
| Fixed charge rate (%) | 7.5 |
| Plant wake loss factor (%) | 15 |

**Table 1.** Preassumed inputs to the financial model computing the LCOE (Stehly and Beiter, 2019).

### 2.1.4  Cost analysis

Throughout the years, the National Renewable Energy Laboratory has released multiple models that estimate the costs of wind energy. This work combines a detailed blade cost model (Bortolotti et al., 2019a), a model to estimate the costs of the other wind turbine components and the overall turbine capital costs (Fingersh et al., 2006), and a financial model to compute the LCOE (Stehly and Beiter, 2019). Although the costs of some components—mostly in the nacelle—are estimated via semiempirical relations tuned on historical data that get updated every few years, the blade cost model adopts a bottom-up approach and estimates the total costs of a blade as the sum of variable and fixed costs. The former is based on materials and direct labor costs, whereas the latter accounts for the costs from overhead, building, tooling, equipment, maintenance, and capital. The blade cost model simulates the whole manufacturing process, and it has been tuned to estimate the costs of blades ranging from 30 to 100 m in length. Last, the financial model from Stehly and Beiter (2019) is updated yearly and computes the LCOE by modeling an entire wind power plant. In this study, the number of turbines, the power losses resulting from turbine wakes and electrical losses, the balance-of-station costs, and the operational expenditures are assumed to be equal to the values reported in Stehly and Beiter (2019) referring to land-based installations. The values are reported in Table 1.

Note that the absolute values of costs contain a wide band of uncertainty, but the authors hope that the models can sufficiently capture the relative trends.

### 2.2  Rail transport of blades

Wind turbine blades are often transported by rail, especially in the United States. Blades are transported rigidly, and the fixtures on the rail cars allow only some amount of lateral sliding. This work removes the assumption of the rigid transport and allows controlled flexing of the blades. The model is described in detail in Carron and Bortolotti (2020) and is briefly summarized here.

The rail transport model simulates the navigation of flatcars through horizontal and vertical curves, ignoring complex scenarios such as S-curves and combinations of horizontal and vertical rail curvatures. The model also assumes very low travel speeds and ignores accelerations besides gravity. The root of the blades is assumed to be located at the center of the first flatcar, and the chord is assumed to be aligned in the vertical direction at zero twist angle, i.e., the trailing edge points upward. The model also assumes that a hydraulic system can move the blade root vertically and can orient the blade axis dynamically during





| Fabric material | Fabric orientation | $E_{11}$ (GPa) | $E_{22}$ (GPa) | G (GPa) | Laminate density (kg m$^{-3}$) | Unit cost ($/kg) |
|---|---|---|---|---|---|---|
| Glass | Unidirectional | 43.7 | 16.5 | 3.3 | 1940 | 1.9 |
| Glass | Biaxial | 11.0 | 11.0 | 13.2 | 1940 | 3.0 |
| Glass | Triaxial | 28.2 | 16.2 | 8.2 | 1940 | 2.9 |
| Carbon | Unidirectional | 157.6 | 9.1 | 4.1 | 1600 | 20.0 |

**Table 2.** Summary of the mechanical properties and unit costs of the composite materials adopted in the blades. Legend—$E_{11}$: longitudinal Young's modulus; $E_{22}$: transversal Young's modulus; and G: shear stiffness modulus.

transport to respect horizontal and vertical clearance profiles, which are assumed to be for oversize loads. In addition, there are hydraulic systems along the blade span that are able to let the blade slide while inducing either flapwise or edgewise bending.

The model computes lateral and vertical reaction forces and computes the wheel flange derailment criterion, which is the ratio between the lateral and vertical forces at the location of the flatcar wheels. Blade strains resulting from bending are considered at the farthest distance from the shear center in the train coordinate system.

Within WISDEM, the rail module is implemented as two sequential nested optimizations: for lateral and for vertical curves. The design variables in the suboptimizations represent the induced loading along the blade span, the angles at the blade root,

and the vertical position of the blade root. The two figures of merit of the two optimization problems are the lateral and vertical reaction forces at the blade root. The optimizations are constrained to keep the blade within the clearance profile for the horizontal and vertical curvatures, to limit the strains along the blade span within a maximum allowable 3,500 microstrains, to maintain at least 10% of the gross weight on each flatcar during navigation of summits, and to avoid exceeding the maximum gross rail weight and prevent damage on the tracks during sags. Once the nested optimizer converges, the ratio of lateral to

vertical forces is estimated and passed to the outer design loop.

## 2.3    Materials

The glass fiber composite materials of the blade are selected from the Montana State University Material Database (Mandell et al., 2016), updated to 2019. Biaxial, triaxial, and unidirectional glass laminates are chosen to be as close as possible to industry standards. An industry-reference pultruded carbon fiber laminate is also adopted from the work of Ennis et al. (2019).

All laminates are assumed to be infused by standard epoxy resin. Table 2 reports a summary of the mechanical properties and unit costs of the laminates used in the rotor blades. Readers interested in the details of the material properties and costs should refer to the section on data availability at the end of this document.

## 2.4    Airfoils

The airfoils of the 10-MW IEA Wind Task 37 reference wind turbine (Bortolotti et al., 2019c) are adopted, adding to the list

the 18% thick airfoil NACA63-618. Table 3 reports the list of airfoils at varying relative thicknesses.



| Name | Relative thickness | Name | Relative thickness |
|------|--------------------|------|--------------------|
| Circle | 100% | SNL-FFA-W3-500 | 50% |
| FFA-W3-360 | 36% | FFA-W3-301 | 30.1% |
| FFA-W3-241 | 24.1% | FFA-W3-211 | 21.1 % |
| NACA63-618 | 18% | | |

**Table 3.** Summary of the airfoils adopted in the blades.

The airfoil polars are computed by running the panel solver XFoil with default settings estimating the Reynolds and Mach numbers at rated conditions. XFoil is run at angles of attack between -10 and 20 deg, and outside of this range a Viterna extension is applied. A correction to account for 3D effects is also applied.

The accuracy of the polars is a known shortcoming of the project so far, and one recommendation for future work consists
of generating new polars with a higher fidelity framework.

### 2.5 Design optimization

This study combines the aerostructural rotor design optimization approach presented in Ning and Petch (2016) with the rail model discussed in Sect. 2.2. The design variables parameterize the blade twist, the chord, the airfoil positions, and the internal thickness of the spar caps at eight stations along the blade span. The design variables at the blade root are kept frozen to avoid
unrealistic models of the blade root, whose diameter is preassumed to be equal to 4.5% of the blade length. The LCOE is set to be the merit figure to be minimized. In addition, inequality constraints limit the chord to the maximum of 4.5 m; impose a minimum margin to stall of 3 deg; limit strains to 3,500 microstrains; impose a margin of at least 10% between the natural frequencies of the blade at the three-per-revolution harmonic of the rotor; and, only for upwind rotors, respect a minimum blade-tower clearance. For the rail transport, an equality constraint imposes that blades have the maximum allowable flapwise
stiffness to successfully navigate horizontal curves with a curvature of 13 deg. This condition guarantees maximum deployment potential in the interior region of the United States and limited deployment potential in other regions. Vertical curves up to radii of 2,000 ft (610 m) are assumed to be representative of the U.S. rail main network (Carron and Bortolotti, 2020) and do not limit the design of 100-m-long blades. In terms of tip-speed ratio, the study assumes the value of 10.5. This allows for reaching the maximum blade tip speed, which is set at 85 m s$^{-1}$, right before rated wind speed. The simple calculation assumes a common
value of the aerodynamic power coefficient of 0.48.

Sequentially after the rotor design optimizations, a tower design optimization is run to design a realistic tower. The tower design optimization aims to minimize tower mass parameterizing outer diameter and wall thickness along tower height. The constraints prevent buckling, violation of ultimate stresses, frequency margins between the first two modes of the tower and the one-per-revolution harmonic of the rotor, and unrealistic ratios of diameter to wall thickness and excessive changes of wall
thickness to ensure the feasibility of the steel hollow cylinders.



## 3 Aeroservoelasticity

Once the optimization converges to a design, the input files of the aeroservoelastic solver OpenFAST are automatically generated (NREL, 2021a). The next subsections discuss the models and assumptions behind this process.

### 3.1 Aerodynamics

OpenFAST implements various models to simulate the aerodynamics of wind turbines (Moriarty and Hansen, 2005). In terms of airfoil aerodynamics, this work adopts the Beddoes-Leishman unsteady model with the Minemma/Pierce variant. The rotor aerodynamics are solved with the combination of blade element momentum theory and the generalized dynamic wake theory, which is used to model skewed and unsteady wake dynamics. Prandtl tip and hub loss models are added to the solutions, whereas the tower wake is modeled via a potential flow model.

Note that these models apply to aerodynamics in the rotor plane and could be inaccurate in the presence of large deflections. The recommendations for future work listed in Sect. 7 suggest a pathway to increase the fidelity level of the rotor aerodynamics.

### 3.2 Rotor elasticity

In this project, the blade elastic properties were modeled in the submodule ElastoDyn, which implements a Euler-Bernoulli beam formulation; and in the submodule BeamDyn (Wang et al., 2017), which models the three blades with a geometrically

exact formulation and requires the six-by-six stiffness and inertia matrices.

PreComp (Bir, 2006) implements a modified classic laminate theory with a shear-flow approach, and it is based on shell elements. PreComp offers the attractive advantages of running almost instantaneously and not requiring sophisticated meshing routines; however, PreComp suffers the limitation that it does not estimate the shear stiffness terms. Notably, in WISDEM Frame3DD is run with no shear and no torsional degree of freedom. In addition, the stiffness and inertia terms computed by

PreComp suffer inaccuracies compared to 3D finite element models (Resor et al., 2010).

A novel open-source framework for the structural analysis of slender composite structures (Feil et al., 2020), named SONATA, was then improved within the BAR project to support the accurate aeroelastic analysis of flexible wind turbine blades. SONATA is adopted here to generate the BeamDyn blade input files. SONATA performs the meshing; performs the 2D and 3D visualizations of the blades; and calls the cross-sectional solvers VABS and ANBA4, which generate identical results

except for off-diagonal stiffness terms related to the precurvature of the beam (Feil et al., 2020).

### 3.3 Controller

The OpenFAST models are controlled via the Reference Open-Source Controller (ROSCO) (Abbas et al., 2021; NREL, 2020). ROSCO implements a standard rotor speed tracking pitch control logic in above-rated wind speeds and an optimal tip-speed ratio tracking control logic in below-rated wind speeds. A wind speed estimator based on an extended Kalman filter reconstructs

the wind speed at hub height to estimate the current tip-speed ratio. A peak-shaving logic limits rotor thrust to 80% of the





nominal value by pitching before rated power is reached. The blades are controlled via a collective pitch system. Finally, a shutdown logic is implemented; it is described in Sect. 3.3.3.

### 3.3.1 Tuning

ROSCO is automatically tuned within WISDEM using the ROSCO generic tuning logic (Abbas et al., 2020). The tuning is
based on tip-speed ratio versus pitch angle tables of power, thrust, and torque coefficients that are generated by running a matrix of cases in CCBlade (Ning, 2014). From the tables, the sensitivity of power to changes in pitch angle and rotor speed is reconstructed. The user provides desired closed-loop response characteristics of the rotor for the pitch and torque control actuator tuning procedure, and the corresponding proportional and integral terms of the controller are calculated automatically.

### 3.3.2 Low pitch rate

A survey among representatives of the wind turbine manufacturing industry returned the concern that very large and flexible wind turbine blades could hit the technological limits of modern pitch systems. This is mainly caused by the large rotational moment of inertia at high deflections. To address this concern, a conservative value of the maximum rate of the pitch actuators is adopted—namely, 2 deg s$^{-1}$. This low pitch rate limits the response time of the rotor to transient events, such as the occurrence of wind gusts and shutdown maneuvers, resulting in higher loads.

### 3.3.3 Shutdown

The standard logic of ROSCO triggers a shutdown when a rotor overspeed or a pitch angle beyond the value corresponding to cut-out wind speed is recorded. When the shutdown is triggered, the desired pitch angle is set to 90 deg, and the pitch systems are actuated at the maximum rate, whereas the torque actuator attempts to slow the rotor speed to zero in a 30-s time window. Notably, the torque actuator does not necessarily saturate, which can cause higher loads than the softer slowing of the rotor that
has been implemented. An additional shutdown trigger has been implemented, where the shutdown procedure is initiated when the filtered signal of the yaw error is higher than 120 deg in average wind speeds greater than 5 m s$^{-1}$. This condition is not common in operational design load cases, but it helps to limit the ultimate loads generated during the occurrence of extreme changes of wind direction.

Downwind rotors do not eliminate the constraint on minimum tower clearance because downwind rotor blades undergo
significant deflections toward the tower in various operating and storm conditions. In this project, the highest deflections have been recorded during emergency shutdown maneuvers below rated wind speed and during operation at wind speeds close to cut out. The shutdown logic of ROSCO for downwind configurations is then adapted in two ways. First, the the cut-out wind speed is reduced from 25 to 19 m s$^{-1}$. Second, the direction of the pitching maneuver for shutdowns triggered by an extreme change of wind direction at and below rated power is switched from 90 to -90 deg and a pitch-to-stall maneuver is initiated. This
operation simultaneously increases lift and drag, and the blades do not flap toward the tower but away from it. A comparison of blade root and tower base bending moments show smaller values than traditional pitch-to-feather shutdown maneuvers.





| | BAR-UAG | BAR-DRG | BAR-DRC | BAR-USC | BAR-URC |
|---|---|---|---|---|---|
| Orientation | Upwind | Downwind | | Upwind | |
| Transport | Air | Rail | | Segmented | Rail |
| Fabric spar caps | Glass | | | Carbon | |
| Prebend at blade tip (m) | 4 | 0 | | 4 | 0 |
| Cone/tilt angles (deg) | 4/6 | 2/5 | 2/5 | 2/4 | 4/8 |
| Cut-out wind speed (m s$^{-1}$) | 25 | 19 | | 25 | |
| Blade mass (tons) | 64.8 | 53.0 | 41.6 | 49.4 | 41.2 |
| Blade cost (k$) | 450 | 407 | 472 | 563 | 466 |
| Turbine capital costs ($ kW$^{-1}$) | 1375 | 1324 | 1343 | 1424 | 1336 |
| Annual energy prod. (GWh) | 23.8 | 23.3 | 23.8 | 24.6 | 24.1 |
| LCOE ($ MWh$^{-1}$) | 44.9 | 44.9 | 44.4 | 45.0 | 43.7 |

**Table 4.** Summary of the turbine designs developed in the study.

Notably, the same operation cannot be performed above rated wind speed because pitching to stall implies crossing the region of angles of attack corresponding to high airfoil efficiencies. Such a procedure would result in higher loads than a conventional pitch-to-feather strategy.

## 4 Rotor designs

Table 4 lists the five designs that have been developed in this study to run analyses and comparisons. The values reported in the table are discussed in the next sections. All designs are three-bladed, have a rated power of 5 MW, a hub height of 140 m, a rotor diameter of 206 m, and a wind class of 3A (IEC-61400-1). Blades do not feature any bend-twist coupling technology, such as sweep and fiber misalignment. All blades have successfully completed the high-fidelity structural design optimization described in Camarena et al. (Under Submission at the Journal of Wind Energy Science).

### 4.1 Upwind – Air transport – Glass fiber spar caps (BAR-UAG)

The first design, which is named BAR-UAG (Upwind – Air transport – Glass fiber spar caps), is an upwind configuration designed without transportation constraints. BAR-UAG blades could be transported to only a limited number of sites or via some advanced air-shipping technology, which is today not commercially available. This design was developed to demonstrate what a rotor design would look like without logistical constraints and to serve as a baseline by which to compare the other BAR designs. The maximum chord is designed for minimum LCOE, and it is equal to 5.3 m. This value exceeds the limit to ensure transportability. Notably, this chord distribution is not the aerodynamically optimal one, but larger values incur marginally higher rotor efficiency and higher blade costs, which are mostly caused by higher building and tooling costs, as estimated by Bortolotti et al. (2019a). The prebend of the blade is limited to 4 m for similar blade manufacturing reasons because higher





values of prebend would require special tooling and special infrastructures, such as higher than usual assembly lines to open and close the blade molds. The blades have spar caps made of glass fiber laminate. The thickness of the spar caps is driven by the constraint on maximum blade tip deflection, and rotor precone and nacelle uptilt angles are set to 4 and 6 deg, respectively, to limit blade mass.

The BAR-UAG design is used to design a reference tower. A design optimization as described in Sect. 2.5 is performed, showing that a standard configuration with a maximum diameter of 4.5 m cannot be designed without violating the constraints, whereas the optimizer converges if the upper bound on the diameters is increased to 6 m. This suggests that—as hinted in the introduction—a novel technological solution such as hybrid concrete-steel or field-welding is needed for the concepts developed in this study.

### 4.2    Downwind – Rail transportable – Glass fiber spar caps (BAR-DRG)

The BAR-DRG wind turbine (Downwind – Rail transportable – Glass fiber spar caps) mounts blades with glass fiber spar caps and meets the rail transportation requirements. Prebend is removed, maximum chord is limited to 4.75 m, and the design optimization significantly reduces the thickness of the blade spar caps to successfully navigate 13-deg horizontal rail curves. The resulting design is much more flexible than the BAR-UAG blades, with savings in blade mass of 18%. The limitation of the BAR-DRG rotor is that it can fly only in a downwind configuration to avoid tower strikes. In terms of power performance, two opposite trends coexist. On one hand, the downwind configuration allows for reducing cone and tilt angles. On the other hand, a poorer power performance is generated because of the reduced swept area caused by the blade deflections. The effects on AEP of the cut out set at 19 m s$^{-1}$ are quantified as equal to 0.2 GWh—namely, 0.9% for the BAR-DRG design. A stiffer rotor design could operate at higher cut out and diminish the power losses, but then it would not comply with the rail transportation requirements.

### 4.3    Downwind – Rail transportable – Carbon fiber spar caps (BAR-DRC)

The BAR-DRC wind turbine (Downwind – Rail transportable – Carbon fiber spar caps) is an evolution of the BAR-DRG design, where the composite of the spar caps is switched from glass to pultruded carbon fiber (Ennis et al., 2019). Thanks to the much higher longitudinal Young's modulus, see Table 2, carbon allows a much thinner relative thickness distribution along the blade span and savings in blade mass of 22% compared to the BAR-DRG blade. The blade cost of BAR-DRC is 16% higher than in BAR-DRG and 5% higher than in BAR-UAG.

### 4.4    Upwind – Segmented – Carbon fiber spar caps (BAR-USC)

The BAR-USC design (Upwind – Segmented – Carbon fiber spar caps) models a segmented blade configuration. The blades, which adopt the baseline carbon fiber material listed in Table 2 in the spar caps, mount a joint at 70% span. The mass and cost of the joint are assumed to be 2,000 kg and $50,000, respectively. Thanks to the carbon spar caps, the blades, which are prebent as in BAR-UAG, are 24% lighter than in BAR-UAG despite a maximum chord limited to 4.75 m. The usage of carbon fiber





| Name | Load | PSF | Description | WS (m/s) | Turb. | Seeds | Gust | Time (s) |
|---|---|---|---|---|---|---|---|---|
| DLC 1.1 | U/F | 1.25 | Normal production | 4:2:19 | NTM | 6 | None | 600 |
| DLC 1.3 | U | 1.35 | Normal production | 4:2:19 | ETM | 6 | None | 600 |
| DLC 1.4 | U | 1.35 | Normal production | $V_r$ +/-2, $V_r$ | None | None | ECD | 100 |
| DLC 1.5 | U | 1.35 | Normal production | 4:2:19 | None | None | EWS | 100 |
| DLC 5.1 | U | 1.35 | Emergency shutdown | $V_r$ +/-2, $V_{out}$ | NTM | 6 | None | 100 |
| DLC 6.1 | U | 1.25 | Parked in extreme wind | $V_{50}$ | NTM | 6 | None | 600 |
| DLC 6.3 | U | 1.25 | Parked with large yaw error | $V_1$ | NTM | 6 | None | 600 |

**Table 5.** Design load cases (DLC) included in this study. Legend—U: ultimate; F: fatigue; PSF: partial safety factor; WS: wind speed; $V_r$: rated wind speed; $V_{out}$: cut-out wind speed; $V_{50}$: 50-year wind speed; $V_1$: 1-year wind speed; NTM: normal turbulence model; ETM: extreme turbulence model; ECD: extreme change of direction; EWS: extreme wind shear. The details of these design load cases are described in IEC-61400-1.

and the presence of the segmentation joint lead to the highest blade costs, namely +25% and +19% compared to the BAR-UAG and BAR-DRC blades. The stiffer spar caps, however, allow for reducing the cone and tilt angles to 2 and 4 deg, respectively, resulting in a higher rotor swept area and the highest AEP among the designs, namely 24.6 GWh.

### 4.5 Upwind – Rail transportable – Carbon fiber spar caps (BAR-URC)

The last design, BAR-URC (Upwind – Rail transportable – Carbon fiber spar caps), is an upwind design with rail-transportable blades made with carbon fiber spar caps. BAR-URC is again straight—i.e., no prebend is present—and the blade is very similar to the BAR-DRC blade. To comply with the tip deflection constraint, the cone and tilt angles of BAR-URC are raised to 4 and 8 deg, respectively. In terms of AEP, despite the significantly higher cone and tilt angles, BAR-URC produces slightly more AEP than BAR-DRC, namely +1.3%. The higher power production is generated by the higher swept area under loading of the 295 upwind design, whereas downwind rotors inherently reduce their swept area when loaded.

## 5 Loads, deflections, and performance

All designs are run through a list of design load cases representative of operational, emergency, and storm conditions. The cases 1.1, 1.3, 1.4, 1.5, 5.1, 6.1, and 6.3 described by the International Electrotechnical Commission (IEC) standards IEC-61400-1 are run for every model, with the turbulent wind modeled at six different occurrences. Table 5 shows the load cases considered 300 for the designs. The simulations are performed in OpenFAST with BeamDyn, and the resulting loads are used by Camarena et al. (Under Submission at the Journal of Wind Energy Science) to run the structural design optimizations. The next sections discuss the aeroservoelastic performance of the designs listed in Table 4.





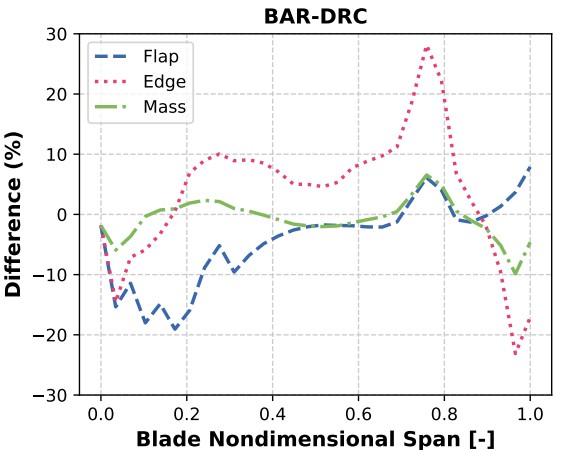
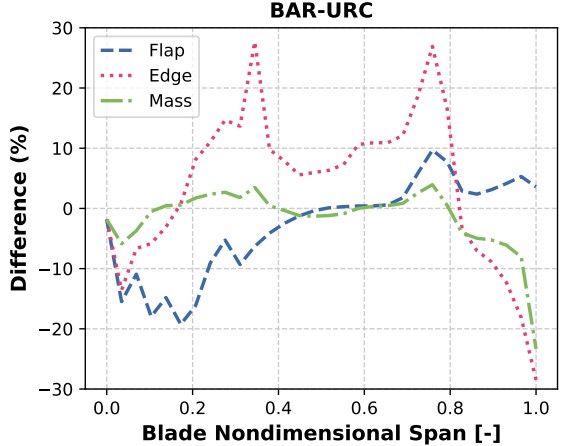

**Figure 2.** Difference in flapwise stiffness, edgewise stiffness, and unit mass distributions between PreComp and ANBA4 for the designs BAR-DRC and BAR-URC. Negative values mean that ANBA4 predicts lower values than PreComp.

## 5.1 Comparison between PreComp and ANBA4

A comparison of the results generated by PreComp and ANBA4 is first run to verify the accuracy of PreComp and to ensure

that the modeling differences between the design process, which is based on PreComp, and the aeroelastic simulations, which are based on ANBA4, are close. The analysis initially considered simple composite beams with a square section, returning minor discrepancies in flapwise and edgewise stiffness and unit mass. Next, the comparison considered the blades described in Sect. 4, showing larger differences. These can be partly attributed to the differences in the meshing logic—i.e., SONATA uses planar elements, whereas PreComp is based on shell elements—and partly to differences between the two solvers.

Figure 2 reports the comparison in terms of flapwise stiffness, edgewise stiffness, and unit mass distributions between PreComp and ANBA4 for the designs BAR-DRC and BAR-URC. The comparison is found somewhat less pessimistic than the one described in Chen et al. (2010), but differences up to ±20% in flapwise and ±30% in edgewise are recorded. The differences in the meshing—which explain the discrepancies between ±5% in the mass distribution—partially explain the differences, but they do not clear them completely. Notably, the BAR-UAG, BAR-DRG, and BAR-USC designs show similar

trends.

The comparison also considered the natural frequencies, which are computed by Frame3DD with the two sets of elastic properties. Here, numbers are closer, with the first five modes within a 3% difference, whereas larger discrepancies are observed for the modes above the fifth. Overall, the comparison concludes that PreComp can be adopted as a conceptual design tool, but higher fidelity structural solvers such as ANBA4 are recommended for a detailed design step.





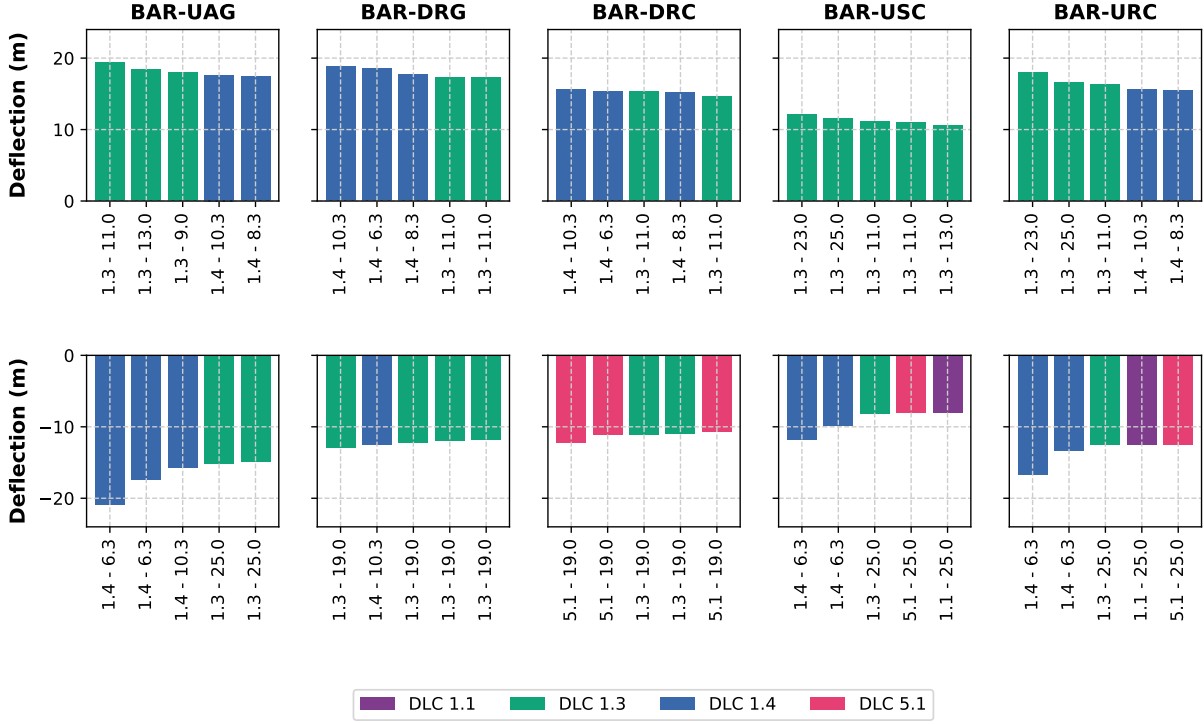

**Figure 3.** Ranking of the out-of-plane blade deflections for the BAR designs. Deflections are aligned with the primary wind direction, so deflections toward the tower are positive in upwind rotors (BAR-UAG, BAR-USC, and BAR-URC) and negative in downwind rotors (BAR-DRG and BAR-DRC). The colors represent the different design load cases (DLCs) listed in Table 5. The deflections do not include partial safety factors.

## 5.2  Loads and deflections ranking

The second step of the analysis consisted of ranking the loads and the blade deflections. The results for the blade deflections toward and away from the tower are shown in Fig. 3. For the upwind designs, the highest out-of-plane blade tip deflections toward the tower are measured during operational load cases at and above rated wind speed—namely, DLC 1.3 at 11 m s⁻¹ for BAR-UAG, DLC 1.3 at 23 m s⁻¹ for BAR-USC, and DLC 1.3 at 23 m s⁻¹ for BAR-URC. Note that BAR-URC has marginally smaller deflections than BAR-UAG, but the nacelle uptilt angle is higher to compensate for the zero prebend, see Table 4. All three designs have a very flat ranking, with several design load cases generating similar ultimate deflections. In terms of deflections away from the tower, all three designs record the highest values during an extreme gust with direction change—namely, DLC 1.4 at 6.3 m s⁻¹.

For the downwind designs, the highest out-of-plane blade tip deflections toward the tower occur during operation or during a shutdown event occurring close to the cut-out wind speed—namely, DLC 1.3 and DLC 5.1 at 19 m s⁻¹. Notably, deflections toward the tower would be higher if the cut-out wind speed were higher. Away from the tower, the highest deflections occur





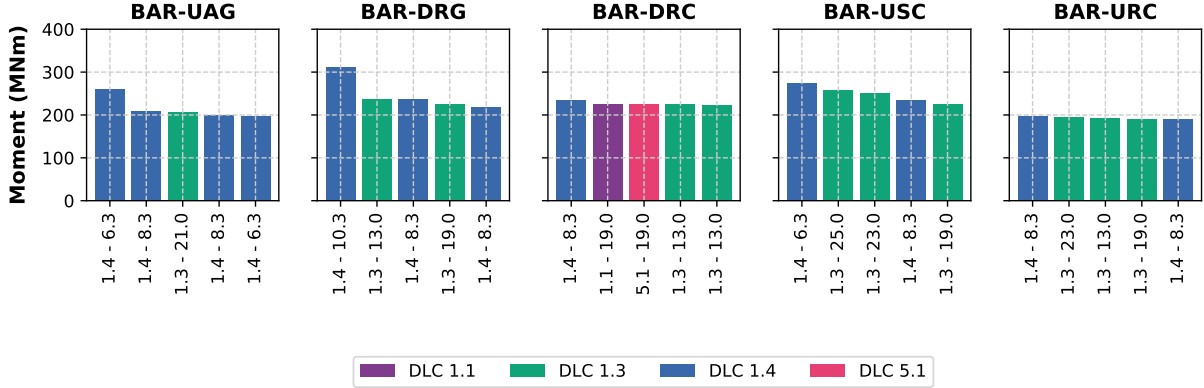

**Figure 4.** Ranking of the fore-aft moments at tower base for the BAR designs. The colors represent the different design load cases (DLCs) listed in Table 5. The moments do not include partial safety factors.

during the extreme gust with direction change, namely DLC 1.4 at 10.3 m s$^{-1}$. BAR-DRG reaches a blade deflection of 19 m, whereas BAR-DRC has a maximum blade deflection of 16 m. These high deflections would cause a tower strike with a standard shutdown maneuver based on a pitch-to-feather strategy; see Sect. 5.3. With the pitch-to-stall strategy, the design

drivers become ultimate loads, as discussed in Camarena et al. (Under Submission at the Journal of Wind Energy Science).

In terms of blade flapwise loads, the rankings follow those of the blade deflections. The ranking of fore-aft tower base moments is reported in Figure 4. DLC 1.4 causes the maximum moments for all designs, with the heavier (BAR-UAG and BAR-DRG) and the most efficient (BAR-USC) designs suffering the highest moments. BAR-URC records the lowest moments, equal to 198 MNm, followed by BAR-DRC at 235 MNm.

**5.3 Shutdown of downwind**

The extreme gust with direction change—namely, DLC 1.4—is found to generate the highest blade deflections and loads; however, the loads and deflections of the flexible downwind rotors BAR-DRG and BAR-DRC would have been higher without the pitch-to-stall strategy applied at and below rated wind speed, as described in Sect. 3.3.3. Figure 5 shows the time histories of the pitch angle, the flapwise tip deflection of Blade 1, the flapwise root moment of Blade 1, and the fore-aft tower base

moment for the pitch-to-feather and pitch-to-stall maneuvers for the BAR-DRC design during an extreme gust with change of direction.

The time histories show lower blade deflection and lower blade and tower moments for the pitch-to-stall maneuver, whereas the pitch-to-feather maneuver leads to a tower strike of Blade 2 at second 25. Note that the this maneuver pushes the angles of attack along the blade span into the deep stall region, which is not very well characterized numerically. Higher fidelity

simulations and field tests are recommended to verify and validate this approach to shutdown.



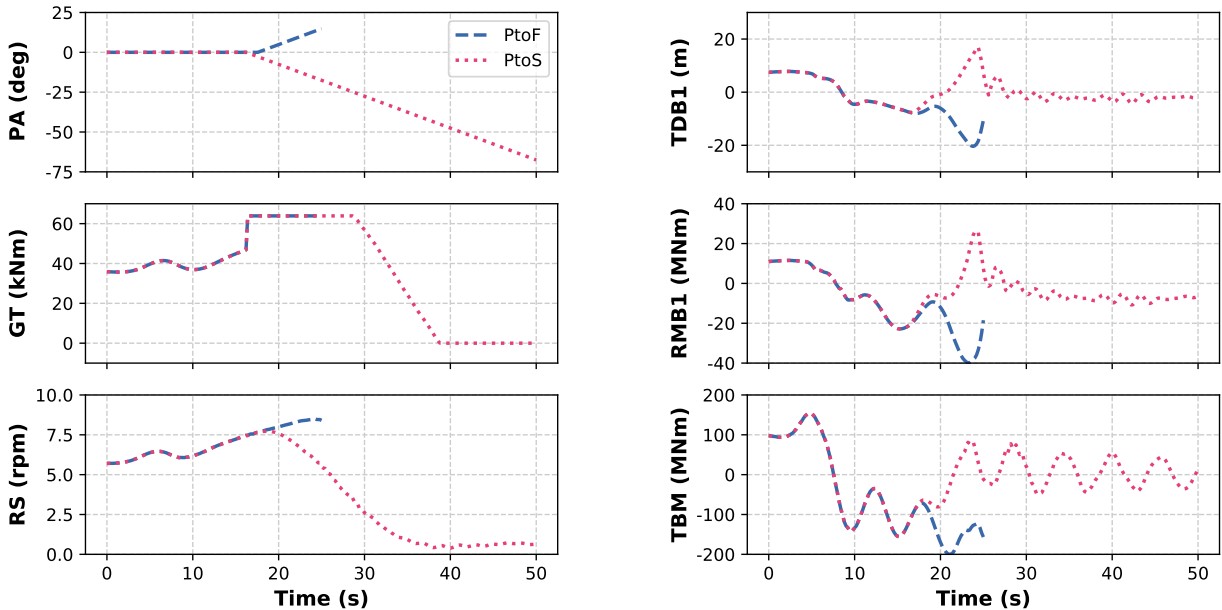

**Figure 5.** Comparison of the time histories of pitch angle (PA) in deg, generator torque (GT) in kNm, rotor speed (RS) in revolutions per minute, flapwise tip deflection of Blade 1 (TDB1) in m, flapwise root moment of Blade 1 (RMB1) in MNm, and fore-aft tower base moment (TBM) in MNm for the pitch-to-feather (PtoF) and pitch-to-stall (PtoS) for the BAR-DRC design during an extreme gust with change of direction. PtoF is interrupted at 25 s because of a tower-strike event of Blade 2.

### 5.4 Torsional deformations and power performance

Increasing the flexibility of the wind turbine rotors leads to increased torsional deformations of the outer blade portions. Figure 6 reports a comparison at varying wind speeds among all five designs at 79% span and at blade tip measured during six occurrences of DLC 1.1. The torsional deformations increase in the rail-transportable designs, especially in the BAR-URC design, which often sees 1 extra degree of torsional rotation compared to the jointed blade BAR-USC. Simplified beam models that do not model the torsional degree of freedom, such as the one implemented in ElastoDyn, are not accurate enough to capture the aeroelastic response of slender wind turbine blades, such as the ones investigated here.

The annual power production is computed by integrating the generator power along six 10-minute simulations with the normal turbulence model at wind speeds spaced by 2 m s$^{-1}$ between cut in and cut out. The power curve is weighted with the Weibull distribution for a Class 3 wind turbine. Note that the results of such an approach are affected by the control tuning, which could slightly vary among designs. The analysis was therefore repeated for steady-state inflow wind conditions, and although the results were different in the absolute values, the trends were found to match.





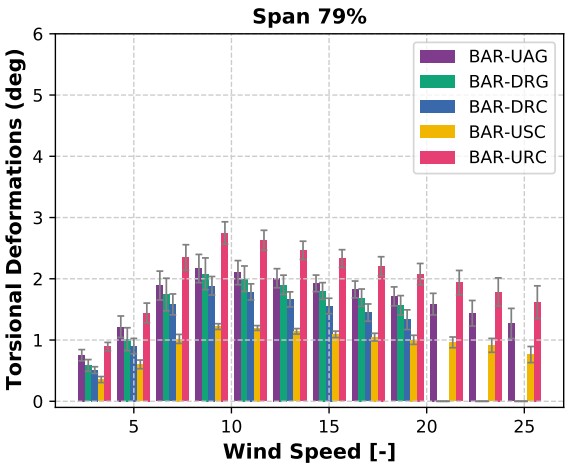 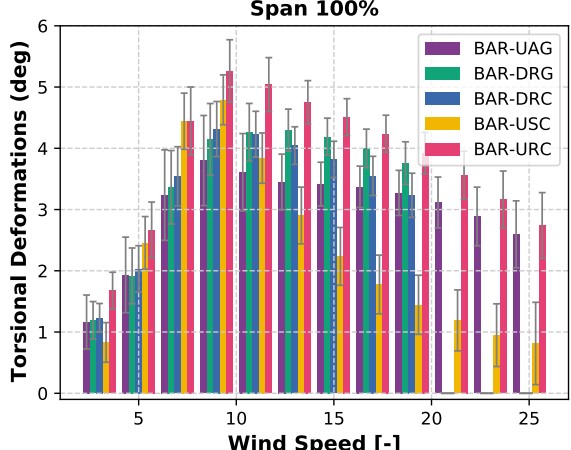

**Figure 6.** Average torsional deformations at 79% blade span and at blade tip at varying wind speeds during six occurrences of DLC 1.1. The error bars indicate the standard deviation.

Despite the high values of torsional deformations, the power performance of the various rotors is not dramatically different; however, two trends are still visible. First, carbon fiber designs support thinner airfoils along the blade span and allow for more
efficient designs. BAR-USC generates 3.4% higher AEP than BAR-UAG, whereas BAR-DRC generates 2.1% higher AEP than BAR-DRG. Second, flexible downwind wind turbines generate less power than equivalent flexible upwind configurations. Qualitatively, this trend can be explained by the lower cut-out wind speed of the downwind rotors and by the fact that when loaded, upwind blades increase the rotor-swept area, whereas the opposite holds true for downwind designs. This trend can be mitigated with stiffer downwind rotors with smaller cone and tilt values, but at equal conditions the LCOE of downwind rotors
is likely higher than upwind. BAR-DRC, although mounting very similar blades and significantly smaller cone and tilt angles, generates 1.2% less AEP than BAR-URC, which in turns generates 2.0% less AEP than BAR-USC.

## 6 LCOE

The study concludes with a discussion about the values listed in Table 4 of turbine capital costs and levelized cost of wind energy of the various designs. All designs are found to generate similar values of LCOE, with BAR-DRC and BAR-URC
generating the best compromise between AEP and turbine capital costs. Compared to BAR-DRG, BAR-DRC compensates the higher blade costs with higher power production generated by the thinner airfoils. The comparison of the values of LCOE returns a 1.1% lower value for the BAR-DRC design compared to the BAR-DRG design. For the upwind configurations, the advantages generated by the lighter BAR-URC rotor lead to a 2.9% savings in LCOE compared to the jointed design BAR-USC. In terms of capital costs, the flexible rotor designs BAR-DRG, BAR-DRC, and BAR-URC achieve the lowest numbers
thanks to low blade costs and system savings generated by the lighter rotors. BAR-DRG reports the lowest value, \$1,324 kW[-1];



closely followed by BAR-DRC and BAR-URC, \$1,343 kW$^{-1}$ and \$1,336 kW$^{-1}$. Note again that these numbers suffer from a wide band of uncertainty and could vary from one manufacturer to another and from one site to another. Also, all designs adopt the same tower, which is designed for BAR-UAG and is likely too heavy and expensive for all other four designs, with the possible exception of BAR-USC. It is clearly shown, however, that highly flexible rotors not only help address logistics

constraints but also can lead to savings in LCOE.

## 7 Conclusions and future work

This work presents the research activities focused on designing novel land-based wind turbines whose blades can be transported by rail via controlled bending. The studies are conducted on a 5-MW platform with a rotor diameter of 206 m and a hub height of 140 m. The activities fall within the BAR project funded by the U.S. Department of Energy and show a promising pathway

toward very flexible wind turbine rotors. The three key conclusions of the work are:

– Controlled bending during rail transportation of both upwind and downwind rotor blades of 100 m in length represents a promising pathway to increasing the size of land-based wind turbines. Longer blades could be adopted—both upwind and downwind—by exceeding conventional values of rotor cone and nacelle uptilt angles.

– Downwind turbines with flexible rotor blades suffer the risk of running into blade-tower strikes; however, blade deflec-

tions toward the tower can be alleviated by reducing the cut-out wind speed and implementing a pitch-to-stall strategy for shutdown maneuvers at and below rated wind speed.

– The use of pultruded carbon fiber in the spar caps leads to slightly more expensive blade designs but also higher aerodynamic performance and lower systems costs, ultimately reducing the LCOE compared to standard glass fabrics.

The recommended activities can be summarized into five main areas:

– The airfoil polars adopted in this study should be replaced with those generated by 2D computational fluid dynamics. Such a framework has only recently become available to the authors and will be used to replace the outputs from XFoil, which can be seen only as a conceptual design tool. The airfoil polars at high Reynolds should also be validated against experimental data. Although this second step would represent a major effort, the lack in the public domain of good quality validation data for airfoil aerodynamics is seen as a strong limitation for the research community.

– The aerodynamics of the flexible rotors should be investigated at higher fidelity, through vortex-based aerodynamic models and 3D computational fluid dynamics. The performance of the highly coned and tilted upwind rotor should also be checked. Validation studies are ongoing with the vortex model implemented in OpenFAST and described in Shaler et al. (2020). The pitch-to-stall shutdown maneuver should finally be a topic of a dedicated investigation.

– Downwind wind turbines have historically suffered from low-frequency noise generation and higher fatigue damage

because of the blades passing in the wake of the tower. Preliminary analysis has increased confidence that the issues



might have affected old wind turbines more than modern configurations (Bertagnolio et al., 2019), but a final answer is
yet to be made.

- The controller of the downwind rotors could be tuned more accurately, exploring new logic to maximize power and
  minimize deflections toward the tower. One possible pathway consists of maintaining a high cut-out wind speed and
  implementing a derating, which would minimize AEP losses while preventing the blade from flying too close to the
  tower. Highly flexible blades could also benefit from distributed aerodynamic control devices more than traditional
  blades. Several research projects are ongoing in this field and should continue.

- This study has adopted WISDEM and OpenFAST, which are characterized by two different levels of fidelities. Work is
  ongoing to adopt formal approaches for the multifidelity optimization of wind turbines.

*Code availability.* All codes used within this work are publicly available. WISDEM is available at https://github.com/WISDEM/WISDEM,
OpenFAST is available at https://github.com/OpenFAST/openfast, and ROSCO is available at https://github.com/NREL/ROSCO. These tools
have recently been integrated into a single framework named WEIS, which can be accessed at https://github.com/WISDEM/WEIS. Last,
SONATA is available at https://gitlab.lrz.de/HTMWTUM/SONATA.

*Data availability.* The input files to WISDEM and WEIS and the OpenFAST models equipped with the ROSCO controller are available at
https://github.com/NREL/BAR_Designs.

*Author contributions.* Pietro Bortolotti has led the wind turbine design process and has led the preparation of this manuscript. Nick Johnson
coleads the Big Adaptive Rotor project and has helped with all tasks. Nikhar Abbas has been in charge of the control development and control
tuning. Evan Anderson and Ernesto Camarena have been leading the detailed structural design of the blades. Joshua Paquette leads the entire
Big Adaptive Rotor project and has provided countless inputs. All six authors have provided valuable feedback and guidance throughout the
entire study.

*Competing interests.* The authors declare that they have no competing interests in executing and publishing this work.

*Acknowledgements.* The authors acknowledge the valuable input provided during the writing of this paper. The authors are grateful to the
whole Big Adaptive Rotor team across the National Renewable Energy Laboratory and Sandia National Laboratories and to the OpenFAST
and the WISDEM teams at the National Renewable Energy Laboratory. The feedback, guidance, and review from all team members are
greatly appreciated. The inputs of the advisory panel throughout the design process of the Big Adaptive Rotor turbines were also extremely
valuable.





A portion of the research was performed using computational resources sponsored by the U.S. Department of Energy's Office of Energy Efficiency and Renewable Energy and located at the National Renewable Energy Laboratory. This work was authored in part by the National Renewable Energy Laboratory, operated by Alliance for Sustainable Energy, LLC, for the U.S. Department of Energy (DOE) under Contract
No. DE-AC36-08GO28308. Funding provided by the U.S. Department of Energy Office of Energy Efficiency and Renewable Energy Wind Energy Technologies Office. The views expressed in the article do not necessarily represent the views of the DOE or the U.S. Government. The U.S. Government retains and the publisher, by accepting the article for publication, acknowledges that the U.S. Government retains a nonexclusive, paid-up, irrevocable, worldwide license to publish or reproduce the published form of this work, or allow others to do so, for U.S. Government purposes.



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
