# Peer review of "Land-based wind turbines with flexible rail transportable blades – Part I: Conceptual design and aeroservoelastic performance"

_Wind Energy Science, 2021_

## Referee Comment (RC1)

[referee-annotated manuscript omitted]

---

## Referee Comment (RC2)

The article presents an interesting investigation with a lot of materials and procedures to deal with the complete problem of LCOE of wind turbines considering the transportation constraints.

All the parts of the study are presented, not in details cause referencing to previous studies or materials, but with sufficient explanations to understand the needs and the specificities of the current study.

The results of the mechanical part are well presented and the trends of each configurations are pointed out. However, the LCOE results are obtained and presented a bit shortly.

A drawback of the article is the way of redaction letting think in the start that the focus will be on the transportation analysis. The reader has to reach the 4th section to understand that the transportation (rail way especially) is only treated has a bending deformation capability. Then it mays be valuable to modify a little the redaction to be clear that the study is a quite classical wind turbine system optimization introducing the flexibility has a new constraint and with an objective function based on the total cost from construction to energy production and also logistic technologies.

The study is based on several previous works and previous stuffs (as softwares) and the writers give a lot of references. Nevertheless, some key points of the mains used tools may be given in order to simplify the reading (it is not a simple task to go check in each reference to get the may assumptions).

A last, the study finally being an optimization problem, it would have been clearer to redact a part as well: design variables, constraints, goal function. And especially, the 4th section presenting the blade configurations may be concluded with a synthetic analysis of the differences between designs. It has to be noticed that the rail way transportation constraint induces design constraints on the blade, that last influencing the optimization of the wind turbine system, but there is no detail of the blade design.

Writing advises:
- There is a lot of "speaking" formulations: penetration (lines 21 and 24), mounts (line 26), aggressive (28), fairly rigid (28), nonnegligible (46) …
- There is "too much" coma: "…chord, prebend, sweep …" (line 33), "…large,jointed, flexible…" (47) …
- The mathematical notation may be avoided in the text: (line 45) "+10-+15%" => "increases from 10% to 15% …"

---

## Author Comment (AC1)

**REVISION TO MANUSCRIPT DRAFT**

**Journal** Wind Energy Science

**Manuscript ID** WES-2021-29

**Title** "Land-based wind turbines with flexible rail transportable blades – Part I: Conceptual design and aeroservoelastic performance"

The authors would like to thank the associate editor Prof. Sandrine Aubrun and the two reviewers Athanasios Barlas and Didier Lemosse for their time and valuable feedback. Their inputs contribute to the improvement of the paper. A list of point-by-point replies to the reviewers' comments is reported in the following. The text from the reviewers is reported in italics and has been divided into a numbered list. Each point is followed by the authors' reply.

**Reviewer #1: Barlas, Athanasios**

1. *[Reviewer] The article presents an interesting and comprehensive study on design optimization of land-based wind turbines with flexible rail transportable blades. Despite possible modelling fidelity limitations, the work is well written and provides good scientific insight in the explored concepts. Detailed comments are provided in the attached pdf file.*

   **[Authors]** We thank the reviewer for his time and valuable feedbacks. We agree with him that the choice of the modeling fidelities is a key aspect of this work. Indeed, it has represented a continuous challenge throughout the first phase of the Big Adaptive Rotor (BAR) project, where major efforts have gone into closing the modeling gaps that existed at the start of the project. In a few cases, these gaps could only be narrowed, and not closed completely. One of these gaps is the fidelity level of the design optimization tools, which have historically relied on lower fidelity codes (not only at NREL). In the past, this choice was made for very good reasons (computational costs, complexity of the framework, availability of higher fidelity tools, etc.), but it represents a growing limitation. Modern highly flexible rotors do require higher fidelity tools for an accurate estimate of their loads and performance. These tools are however not readily available in a context of numerical design optimization, they have significantly higher computational costs, and often do not guarantee a smooth solution space. Overall, running numerical design optimization at higher fidelity is not trivial.
   In this work, we try to overcome this challenge with a two-step iterative approach, where we first run the design scripts at lower fidelity, we evaluate the resulting turbine designs at higher fidelity, and we then manually iterate updating the constraints at the design lever. This approach seems to work well in practice, but we agree that a higher degree of automation will be beneficial. The path that we envision is to integrate more and more the higher fidelity tools within the design loops. The newly developed NREL code WEIS (https://github.com/WISDEM/WEIS) goes in this direction and we hope to generate new results with it soon. Future activities will also strengthen the integration between classical aeroservoelastic tools and the latest frameworks for blade-resolved computational-fluid-dynamics. The recommendations for future work listed in this journal article trace these considerations.

2. *[Reviewer] Page 1 Line 11 - A 'segmented design' is firstly mentioned here, so it's not clear for the abstract. Please consider revising or describing it above.*

   **[Authors]** Good catch, we added a sentence introducing the segmented design earlier in the abstract.

3. *[Reviewer] Page 2 Line 23 - I believe the term 'penetration', used to describe the contribution/share of renewables in the energy mix, is avoided lately for political reasons.*

   **[Authors]** We removed the word "penetration".

4. *[Reviewer] Page 2 Line 55 - Consider summarizing the 5 designs with bullet points or in a table, since this paragraph could be hard to follow.*

**[Authors]** We modified the paragraph to improve its readability. Table 4 summarizes the key features of the five designs, and we suggest leaving it in Section 4.

5. ***[Reviewer]*** *Page 4 Line 90 - Although the choice of the simple aero loading tool is obvious from the MDAO point of view, it is not easily justified considering the complexity of the aeroelastic response of highly flexible and coned rotors. Moreover, this choice is not justified by any of the provided references in this section. Please comment.*

   **[Authors]** See comment #1. This aspect was listed in the recommendations for future work but was possibly not well discussed in the rest of the paper. We restructured the last few paragraphs of the introduction to better motivate the choices behind our work.

6. ***[Reviewer]*** *Page 7 Line 151 - Are the polars computed for free boundary layer transition, or is a mix with fully turbulent polars utilized? Please comment and justify the choice.*

   **[Authors]** The polars were computed using the standard settings of XFoil, which correspond to free boundary layer transition. The choice was dictated by the simple fact that XFoil was the only tool that we had available at the beginning of the BAR project to compute polars for airfoil thickness between 18% and 50%. As the text already states, this is a known shortcoming of our analysis, and we are working to generate a mix of free transition and fully turbulent polars with a 2D CFD framework described in this publication DOI [10.5194/wes-2021-23](https://doi.org/10.5194/wes-2021-23). However, this activity has its own set of challenges, and will be described in a future publication. We slightly adapted the text to make sure that readers understand the choices and challenges behind the polars.

7. ***[Reviewer]*** *Page 7 Line 157 - What about control settings (rpm, pitch), in/out-of-plane positioning of aero sections and spar cap location? Are control settings only tuned later in the FAST simulations?*

   **[Authors]** The control settings are defined by a regulation trajectory, as briefly introduced in Section 2.1.1. The spar cap locations were manually chosen to have the spar caps centered around the chordwise point of maximum airfoil thickness. Their width was optimized in NuMAD. The chordwise airfoil positions were manually set to have smooth leading and trailing edges and to minimize the offset between the axis going through the rotation centers and the sectional centers of gravity. Finally, the damping and frequency for pitch and torque actuators in ROSCO/OpenFAST were set based on studies conducted over the IEA Wind Task 37 reference wind turbines. Overall, the design optimization is currently capable of tuning only a subset of parameters, and we had to adopt several assumptions over the various design steps. The paper now includes a note about this and about the considerations above.

8. ***[Reviewer]*** *Page 8 Line 183 - That's probably not accurate for the downwind configurations, where a jet model is more appropriate.*

   **[Authors]** This an interesting comment and we were not aware of this aspect. The tower wake is however one of those modeling gaps that has not been closed completely, and uncertainties about the impact of the tower wake on downwind rotors is still an open research question. We will look into it in the second three-year phase of the BAR project.

9. *[Reviewer] Page 8 Line 186 - That's a very good remark. Especially given the fact that the optimizer is utilizing an even simpler BEM models to converge to optimal designs. Some important design trends might be missing in that process for highly flexible and downwind rotors.*

    **[Authors]** As discussed in comments #1 and #5, we totally agree that this is an area of research that deserves more efforts. The combination of the new text in the introduction and this note supports the recommendation provided in Section 7.

10. *[Reviewer] Page 8 Line 203 - Are any optimized pitch setting used in below rated region?*

    **[Authors]** No, but we have added a note about this in Sect. 3.3. Note that the blades presented in this paper do not mount any bend-twist coupling feature and therefore the torsional deformation below rated wind speed are somewhat limited. Still, we agree with the reviewer that this could be something to investigate, and we included a recommendation for future work to tune the controller with the full aeroelastic model of the blade.

11. *[Reviewer] Page 12 Table 5 - Yaw angles? Seeds per yaw angle?*

    **[Authors]** Thank you for pointing out that Table 5 needed some attention. Among various other changes, we now specify the yaw angles of +/- 20 degrees.

12. *[Reviewer] Page 12 Table 5 - DLC1.1 is normal used for statistical extrapolation and DLC1.2 for fatigue analysis, and it's not clear if that's the case here.*

    **[Authors]** The aeroelastic simulations of land-based wind turbines needed to run DLC 1.1 and 1.2 are identical, and only the post-processing of the outputs differs. The analysis of the ultimate loads was included in this work, see for instance Figure 3, Figure 4, and Figure 6, whereas the fatigue loads from DLC 1.2 were only used in the paper Part 2 from Camarena et al., 2021. This said, Table 5 was confusing, so we followed your suggestion and we added one row for DLC 1.1 and one row for DLC 1.2.

13. *[Reviewer] Page 12 Line 297 - Any justification for the reduced DLB?*

    **[Authors]** Unfortunately we do not have a very strong justification, beyond the fact that the subset already includes more than 200 simulations and that the cases are similar to the design load cases adopted by other publications, such as DOIs 10.1002/we.2270 and 10.2172/1529216. Qualitatively, the list covers the most known cases, i.e. DLC 1.x and 6x, with the addition of DLC 5.1 that is relevant for downwind rotors because of the snapback of the blades toward the tower during the shutdown maneuver. We added a note highlighting that the list is indeed a subset and that the results might suffer from this assumption.

1. ***[Reviewer]*** *The presented study deals with the "complete" life of a wind turbine system, from construction and transportation to energy production. That study then introduced of lot of analysis processes and a lot of dedicated tools (blade design, airfoils, optimization patterns, coupled aero-servo-elastic simulations and so on). Introducing the transportation of the blade from the construction factory to the wind turbines farm is a real challenge and may be treated in several ways. Here the authors want to evaluate the rail way transportation situation. This is a coupled problem because the transporation mode induced design constraints on the blades and then modify the whole system behaviour. It is an really interesting and valuable contribution for the wind turbine domain development. The paper is more presented as an application study than as a scientific analysis, but nevertheless need to mastering a lot of scientific concepts alltogether. The introduction part of the article could be modified in order to make its topic more obvious. More details in the added document.*

   **[Authors]** We would like to thank the reviewer for his time, for the valuable feedbacks, and for acknowledging the quality of the work. Following his comments #1 and #3, we have slightly adjusted the introduction.

2. ***[Reviewer]*** *The article presents an interesting investigation with a lot of materials and procedures to deal with the complete problem of LCOE of wind turbines considering the transportation constraints. All the parts of the study are presented, not in details cause referencing to previous studies or materials, but with sufficient explanations to understand the needs and the specificities of the current study. The results of the mechanical part are well presented and the trends of each configurations are pointed out. However, the LCOE results are obtained and presented a bit shortly.*

   **[Authors]** We agree with the reviewer that the paragraph about the LCOE analysis is a little short. This was a deliberate choice dictated by the vast uncertainties that affect the LCOE metrics to let the reader focus on the metrics such as masses, loads, deflections, and performance, which are more certain than costs. Work is ongoing within the second phase of the BAR project to expand the LCOE analysis and challenge some of the assumptions that have so far been taken. This said, in the first paper submission, Section 6 was not balanced compared to Section 5 and we have now merged the two sections. This hopefully improves the readability of the document better guiding the reader toward the key takeaways of this work.

3. ***[Reviewer]*** *A drawback of the article is the way of redaction letting think in the start that the focus will be on the transportation analysis. The reader has to reach the 4th section to understand that the transportation (rail way especially) is only treated has a bending deformation capability. Then it mays be valuable to modify a little the redaction to be clear that the study is a quite classical wind turbine system optimization introducing the flexibility has a new constraint and with an objective function based on the total cost from construction to energy production and also logistic technologies.*

   **[Authors]** This is a great suggestion that we have now incorporated by changing this sentence "This work aims to investigate an alternative pathway—namely, the controlled flexing during rail transport of monolithic blades." to "This work aims to investigate an

alternative pathway---namely, the system design of wind turbines whose monolithic blades can be transported on rail via controlled flexing.". As for the sophistication of the transportation model, which is described in Sect. 2.2 and in DOI [10.1088/1742-6596/1618/4/042041](10.1088/1742-6596/1618/4/042041), it is true that it is low fidelity, but it is also somewhat more than a simple constraint on bending stiffness. The blades that meet the rail transportation constraints are indeed straight, i.e. no prebend nor sweep, and have a maximum chord of 4.75 meters. The rail transport model also checks the strains during bending, although they are lower than in operation and they are therefore not active during the design process.

4. **[Reviewer]** *The study is based on several previous works and previous stuffs (as softwares) and the writers give a lot of references. Nevertheless, some key points of the mains used tools may be given in order to simplify the reading (it is not a simple task to go check in each reference to get the may assumptions).*

   **[Authors]** We agree with the reviewer that many technical details have been left to the references, and that a thorough reader could easily spend more time going through the bibliography than the actual text. Nonetheless, during the writing our goal has always been to highlight the key takeaways of this work and focus on the novelty aspects. To achieve this goal, we wanted to keep the article to a manageable size, and so the heavy use of references.

5. **[Reviewer]** *A last, the study finally being an optimization problem, it would have been clearer to redact a part as well: design variables, constraints, goal function. And especially, the 4th section presenting the blade configurations may be concluded with a synthetic analysis of the differences between designs. It has to be noticed that the rail way transportation constraint induces design constraints on the blade, that last influencing the optimization of the wind turbine system, but there is no detail of the blade design.*

   **[Authors]** Section 2.5 discusses the setup of the design optimizations in terms of design variables, constraints, and figure of merit. The results in terms of blade design are presented only in terms of aggregate blade mass and blade cost. As indicated in the section "Data availability", all codes and wind turbine models are publicly available. We added a note that the models are also available in the wind turbine ontology format windIO (https://windio.readthedocs.io/en/latest/) defined within the IEA Wind Task 37. We also decided to prepare a technical report that describes the resulting blade designs in more detail. We added a reference to this document at the beginning of Section 4.

**Writing advises**

6. **[Reviewer]** *There is a lot of "speaking" formulations: penetration (lines 21 and 24), mounts (line 26), aggressive (28), fairly rigid (28), nonnegligible (46) …*

   **[Authors]** We removed the word "penetration", and we removed the words "fairly" and "nonnegligible". We instead propose to keep using the word "mount".

7. **[Reviewer]** There is "too much" coma: "…chord, prebend, sweep …" (line 33), "…large,jointed, flexible…" (47) … The mathematical notation may be avoided in the text: (line 45) "+10-+15%" =>"increases from 10% to 15% …"

**[Authors]** Thank you for these suggestions. This paper has been reviewed by the editors of the communication department of the National Renewable Energy Laboratory. They have a vast experience in redacting and editing scientific publications, and we followed their guidelines. We would therefore propose to keep these styling choices as they are.